

# 1 Prior biosphere model impact on global terrestrial CO₂ fluxes estimated
# 2 from OCO-2 retrievals

Sajeev Philip[1,2], Matthew S. Johnson[1], Christopher Potter[1], Vanessa Genovesse[3,1], David F. Baker[4,5],
Katherine D. Haynes[6], Daven K. Henze[7], Junjie Liu[8], and Benjamin Poulter[9]
[1]NASA Ames Research Center, Moffett Field, CA 94035, USA
[2]NASA Postdoctoral Program administered by Universities Space Research Association, Columbia, MD 21046, USA
[3]California State University, Monterey Bay, CA 93955, USA
[4]NOAA Earth System Research Laboratory, Global Monitoring Division, Boulder, CO 80305-3337, USA
[5]Cooperative Institute for Research in the Atmosphere, Colorado State University, Ft. Collins, CO 80521, USA
[6]Department of Atmospheric Science, Colorado State University, Fort Collins, CO 80523, USA
[7]Department of Mechanical Engineering, University of Colorado at Boulder, Boulder, CO 80309, USA
[8]Jet Propulsion Laboratory, California Institute of Technology, Pasadena, CA 91109, USA
[9]NASA Goddard Space Flight Center, Greenbelt, MD 20771, USA
*Correspondence to*: Sajeev Philip (philip.sajeev@gmail.com) and Matthew S. Johnson (matthew.s.johnson@nasa.gov)
**Abstract.** This study assesses the impact of different state-of-the-science global biospheric CO₂ flux models, when applied as prior
information, on inverse modeling "top-down" estimates of terrestrial CO₂ fluxes obtained when assimilating Orbiting Carbon
Observatory 2 (OCO-2) observations. This is done with a series of Observing System Simulation Experiments (OSSEs) using
synthetic CO₂ column-average dry air mole fraction (XCO₂) retrievals sampled at the OCO-2 satellite spatio-temporal frequency.
The OSSEs used the four-dimensional variational (4D-Var) assimilation system with the GEOS-Chem global chemical transport
model (CTM) to estimate CO₂ net ecosystem exchange (NEE) fluxes using synthetic OCO-2 observations. The impact of biosphere
models in inverse model estimates of NEE is quantified by conducting OSSEs using the NASA-CASA, CASA-GFED, SiB-4 and
LPJ models as prior estimates and using NEE from the multi-model ensemble mean of the Multiscale Synthesis and Terrestrial
Model Intercomparison Project as the "truth". Results show that the assimilation of simulated XCO₂ retrievals at OCO-2 observing
modes over land results in posterior NEE estimates which generally reproduce "true" NEE globally and over terrestrial TransCom-
3 regions that are well-sampled. However, we find larger spread among posterior NEE estimates, when using different prior NEE
fluxes, in regions and seasons that have limited OCO-2 observational coverage and a large range in "bottom-up" NEE fluxes.
Posterior NEE estimates had seasonally-averaged posterior NEE standard deviation (SD) of ~10% to ~50% of the multi-model-
mean NEE for different TransCom-3 land regions with significant NEE fluxes (regions/seasons with a NEE flux ≥ 0.5 PgC yr⁻¹).
On a global average, the seasonally-averaged residual impact of the prior model NEE assumption on posterior NEE spread is ~10-
20% of the posterior NEE mean. Additional OCO-2 OSSE simulations demonstrate that posterior NEE estimates are also sensitive
to the assumed prior NEE flux uncertainty statistics, with spread in posterior NEE estimates similar to those when using variable
prior model NEE fluxes. In fact, the sensitivity of posterior NEE estimates to prior error statistics was larger compared to prior
flux values in some regions/times of the Tropics and Southern Hemisphere where sufficient OCO-2 data was available and large
differences between the prior and "truth" were evident. Overall, even with the availability of dense OCO-2 data, noticeable residual
differences (up to ~20-30% globally and 50% regionally) in posterior NEE flux estimates remain that were caused by the choice
of prior model flux values and the specification of prior flux uncertainties.





## 1 Introduction

Carbon dioxide ($CO_2$) is the most important greenhouse gas (GHG) contributing to climate change on a global scale (IPCC, 2014). The anthropogenic emission of $CO_2$, primarily from fossil fuel usage, has led to average global $CO_2$ mixing ratios reaching historically high levels of > 400 parts per million (ppm) (Seinfeld and Pandis, 2016). In addition to fossil fuel emissions, the processes involved in the exchange of carbon between the atmosphere and terrestrial biosphere are a major factor controlling atmospheric concentrations of $CO_2$ (e.g., Schimel et al., 2001) with an estimated global biosphere sink of ~3.0 PgC yr$^{-1}$ (Le Quéré et al., 2018). However, current estimates of regional-scale atmosphere-terrestrial biosphere $CO_2$ exchange have large uncertainties (Schimel et al., 2015). "Bottom-up" techniques typically simulate the atmosphere-terrestrial biosphere exchange based on our understanding of these complex exchange processes and by constraining these estimates with remote-sensing inputs and limited measurements available for evaluation. Previous studies inter-comparing several of the most commonly used biospheric flux models (Heimann et al., 1998, Huntzinger et al., 2012; Sitch et al., 2015; Ott et al., 2015; Ito et al., 2016) and multi-model ensemble integration projects (Schwalm et al., 2015) reveal a large spread among global/regional "bottom-up" terrestrial biospheric flux estimates and the sub-components such as ecosystem primary production and respiration (Huntzinger et al., 2012).

An alternate approach to estimate biospheric $CO_2$ fluxes is through "top-down" estimation techniques using inverse models with highly accurate in situ data (e.g., Baker et al., 2006b) or dense and globally distributed satellite data (e.g., Chevallier et al., 2005). The Orbiting Carbon Observatory-2 (OCO-2) satellite, launched in 2014, is the space-borne sensor with the finest resolution and highest sensitivity of $CO_2$ in atmospheric boundary layer to date (Crisp et al., 2017; Eldering et al., 2017a). Studies applying OCO-2 retrievals revealed the ability to investigate novel aspects of the carbon cycle (e.g., Eldering et al., 2017b; Liu et al., 2017), however, the "top-down" estimates of surface $CO_2$ fluxes from numerous inverse modeling systems, using identical OCO-2 observations, show differences among optimized/posterior regional $CO_2$ fluxes (Crowell et al., in prep). Previous studies investigating $CO_2$ flux inversions (e.g., Peylin et al., 2013; Chevallier et al., 2014; Houweling et al., 2015) suggest that this spread among optimized $CO_2$ flux estimates could be due to numerous factors, such as the accuracy and precision of observation data (Rödenbeck et al., 2006), imperfect observation coverage (Liu et al., 2014; Byrne et al., 2017), data density (Law et al., 2003; Rödenbeck et al., 2003) and poorly characterized measurement error covariance (Law et al., 2003; Takagi et al., 2014). Variations in inverse estimation setups between modeling groups, such as model transport (Chevallier et al., 2010; Houweling et al., 2010; Basu et al., 2018) and inversion methods (Chevallier et al., 2014; Houweling et al., 2015), could also lead to inter-model spread in posterior estimates.

In addition to the variables listed above, the assumed prior fluxes and the associated prior error covariance can also impact "top-down" global/regional $CO_2$ flux estimates (e.g., Gurney et al., 2003). Gurney et al. (2003) assessed the sensitivity of $CO_2$ flux inversions to the specification of prior flux uncertainty and found that the posterior estimates were sensitive to the prior fluxes over regions with limited in situ observations. In addition, Wang et al. (2018) found that optimal $CO_2$ flux allocation over land versus ocean, using satellite and/or in situ data assimilations, is sensitive to the specification of prior flux uncertainty. Furthermore, Chevalier et al. (2005) and Baker et al. (2006a; 2010) highlighted the importance of accurate assumptions of prior flux uncertainty by conducting four-dimensional variational (4D-Var) assimilation of satellite column retrievals of $CO_2$. However, to date, there are no controlled experimental studies to isolate and quantitatively assess the impact of assumed prior fluxes and prior uncertainty to inverse estimates of biospheric $CO_2$ fluxes using satellite observations.

Therefore, during this study we conduct a series of controlled experiments to quantitatively assess the impact of assumed prior fluxes and prior uncertainty on global and regional $CO_2$ inverse model flux estimates when assimilating OCO-2 data. In order to achieve this, a series of Observing System Simulation Experiments (OSSEs) are conducted using synthetic OCO-2 observations in the GEOS-Chem 4D-Var assimilation system, with four different prior "bottom-up" NEE $CO_2$ flux estimates. Section 2 of this



study describes the methods applied during this work including models and model input, synthetic OCO-2 data and the inversion
technique applied in the OSSEs. Section 3 presents the forward and inverse model results of simulated atmospheric $CO_2$
concentrations and inferred posterior flux estimates. Finally, our concluding remarks and discussion are described in Section 4.
**2 Methods**
To quantify the impact of prior model NEE predictions on posterior estimates of biospheric $CO_2$ fluxes, a series of $CO_2$ forward
and inverse model simulations were conducted with four different state-of-the-science biosphere models. OSSE simulations were
designed to isolate the differences in posterior NEE estimates caused by the selection of prior model biospheric $CO_2$ fluxes and
uncertainties when assimilating OCO-2 observations. The OSSE framework, input variables, inversion technique and analysis
method are presented below.
**2.1 Prior NEE fluxes**
NEE is the net difference of gross primary production (GPP) and total ecosystem respiration ($R_e$), which itself is the sum of
autotrophic respiration ($R_a$) and heterotrophic respiration ($R_h$). NEE, estimated by terrestrial biospheric $CO_2$ flux models, is
commonly applied in CTMs to simulate atmosphere-terrestrial biosphere carbon exchange. Many biosphere carbon models
estimate GPP and $R_e$, however, some models simulate net primary productivity (NPP), which is defined as the difference between
GPP and $R_a$. In this study, we apply year-specific NEE fluxes calculated from four state-of-the-science biosphere models: 1) NASA
Carnegie Ames Stanford Approach (NASA-CASA), 2) CASA-Global Fire Emissions Database (CASA-GFED), 3) Simple-
Biosphere model version 4 (SiB-4) and 4) Lund-Potsdam-Jena (LPJ). It should be noted that the prior biosphere models used in
this study include only NEE and a single dataset for wild fire and fuel wood burning $CO_2$ emissions was added separately (see
Sect. 2.3). The models applied during this study represent a range of diagnostic approaches, from models predicting biospheric
$CO_2$ fluxes using remotely-sensed data (e.g., Fraction of Absorbed Photosynthetically Active Radiation, Leaf Area Index,
Normalized Difference Vegetation Index) to fully prognostic models unconstrained by observations. In addition, we selected both
balanced/neutral (SiB-4) and non-balanced (NASA-CASA, CASA-GFED, LPJ) biospheric fluxes in our OSSEs in order to
represent the range of prior models currently being used in $CO_2$ inversion modeling studies.
CASA is an ecosystem model predicting NPP based on light use efficiency and $R_h$ based on soils/plant production
information (Potter et al., 1993; 2012b). The NASA-CASA model is a version of the original CASA model (Potter et al., 1993)
currently being developed at NASA Ames Research Center (Potter et al., 2003; 2007; 2009; 2012a; 2012b). NASA-CASA
specifically utilizes data on global vegetation cover (enhanced vegetation index, surface solar irradiance data) and land disturbances
retrieved from the NASA Moderate Resolution Imaging Spectroradiometer (MODIS) satellite (Potter et al., 2012b). In addition to
$R_h$, NASA-CASA includes redistributed crop harvest $CO_2$ emissions to the atmosphere (Potter et al., 2012b). The CASA-GFED
model is a different version of the original CASA model and is described in Randerson et al. (1996) with subsequent versions
being described in recent literature (van der Werf et al., 2004; 2006; 2010). NASA-CASA and CASA-GFED differ in the use of
input parameters and some of the parameterizations (see Ott et al. (2015) for further description).
The SiB-4 model was developed at Colorado State University (Sellers et al., 1986; Denning et al., 1996) with details of
the newest versions described in Haynes et al. (2013). This model is a mechanistic, prognostic land surface model that integrates
heterogeneous land cover, environmentally responsive prognostic phenology, dynamic carbon allocation and cascading carbon
pools from live biomass to surface litter and soil organic matter (Haynes et al., 2013; Baker et al., 2013; Lokupitiya et al., 2009;
Schaefer et al., 2008; Sellers et al., 1996). By combining biogeochemical, biophysical and phenological processes, SiB-4 predicts
vegetation and soil moisture states, land surface energy and water budgets and the terrestrial carbon cycle. Rather than relying on



satellite input data, SiB-4 fully simulates the terrestrial carbon cycle by using the carbon fluxes to determine the above and below-
ground biomass, which in turn feed back to impact carbon assimilation and respiration. Similar to NASA-CASA, the SiB4 model
redistributes crop harvest $CO_2$ emission to the atmosphere. Note that we use a balanced (neutral) biospheric NEE flux (balanced
biosphere for the 1998-2017 time period) for the SiB-4 model.

5          The LPJ model is a process-based dynamic global vegetation model (Sitch et al., 2003; Polter et al., 2014). The LPJ-wsl

dynamic global vegetation model (Sitch et al., 2003) was used to simulate NEE using meteorological data from the Climate
Research Unit (Harris et al., 2013). LPJ is fully prognostic, meaning that the establishment, growth and mortality of vegetation are
represented by first-order physiological principles. The model includes nine plant functional types distinguished by their
phenology, photosynthetic pathway and physiognomy. Phenology status is determined daily and photosynthesis is estimated using
a modified Farquhar scheme (Haxeltine and Prentice, 1996). NPP is calculated from photosynthesis after accounting for $R_a$ and
reproductive allocation. The LPJ-wsl model has been evaluated in several benchmarking activities for stocks and fluxes (Peng et
al., 2015; Sitch et al., 2015).

13          In order to provide a "true" NEE flux for the OSSEs conducted in this study (Sect. 2.4), we use the multi-model ensemble

NEE mean from the Multiscale Synthesis and Terrestrial Model Intercomparison Project (MsTMIP) (Huntzinger et al., 2013; 2018;
Fisher et al., 2016a; 2016b). The MsTMIP NEE fluxes are from a weighted ensemble mean of 15 biosphere models (Schwalm et
al., 2015) for the year 2010. Here we apply the MsTMIP data for year 2010 as the "truth" with year-specific prior model predictions
for 2015. This procedure is justified in our case as within an OSSE framework there needs to be a difference between "true" and
prior fluxes, as long as the "true" values are realistic in nature. The MsTMIP ensemble NEE mean represents a summary over all
15 models which smooths out errors particular to any given model. This "true" NEE flux is used to produce the synthetic OCO-2
observations applied in this study (described further in Sect. 2.4.3).

21          The four prior and the "true" NEE fluxes were regridded from their native horizontal resolutions to the grid resolution of

the inverse model simulations (4.0° latitude × 5.0° longitude). The MsTMIP NEE fluxes are provided as 3-hourly averages and
the four year-specific prior models were provided as monthly-mean GPP or NPP and $R_e$ or $R_h$. Therefore, we imposed diurnal
(hourly) and daily variability to these four prior models following the approach in CarbonTracker CT2016
(http://carbontracker.noaa.gov), which is based on Olsen and Randerson (2004). This hourly/daily NEE variability for each prior
model was calculated using the downward solar radiation flux and 2-meter air temperature data from the GEOS-FP (Goddard Earth
Observing System Model, Version 5 "Forward Processing") meteorological product and monthly-averaged GPP and $R_h$ from the
respective model. We allow the "true" and prior models to have different diurnal variability in order to represent a realistic scenario,
as prior models will differ some from the actual diurnal variability of NEE in nature. Table 1 shows the global annual NEE flux
estimates for the four prior models and the "truth". From this table it can be seen that the MsTMIP product shows a strong annual
global sink of -4.31 PgC yr⁻¹. NASA-CASA and CASA-GFED predict a global sink of ~2 PgC yr⁻¹ and differ by ~0.6 PgC yr⁻¹.
SiB-4 NEE predicts a source of ~1 PgC yr⁻¹ and the LPJ model predicts a strong sink of ~5.5 PgC yr⁻¹. Section 3.1 further describes
the spatio-temporal differences of the NEE fluxes between these four prior models and the "truth".
**2.2 GEOS-Chem model**
The GEOS-Chem chemical transport model (CTM) (http://geos-chem.org; Bey et al., 2001) used in this study has the capability to
run forward $CO_2$ simulations (Suntharalingam et al., 2004; Nassar et al., 2010) and corresponding adjoint model calculations
(Henze et al., 2007; Liu et al., 2014). In this study, we use the GEOS-Chem adjoint version 35, which is compatible with version
8-02 of the GEOS-Chem forward model. Liu et al. (2014) tested the accuracy of the GEOS-Chem $CO_2$ adjoint system, which has
been used for several $CO_2$ inverse modeling studies (e.g., Liu et al., 2017; Bowman et al., 2017; Deng et al., 2014). The model is



driven with assimilated meteorological fields from the GEOS-FP model of the NASA Global Modeling Assimilation Office
(GMAO). The GEOS-FP meteorology fields have a native horizontal resolution of $0.25° \times 0.3125°$ and 72 native hybrid sigma-
pressure vertical levels from the Earth's surface to 0.01 hPa. We conduct simulations with a coarser spatial resolution ($4.0° \times 5.0°$)
with 47 reduced vertical levels to attain reasonable computational efficiency.
**2.3 Non-NEE $CO_2$ fluxes**
To simulate concentrations of atmospheric $CO_2$, we used several land and ocean $CO_2$ flux inventories in addition to the NEE
estimates from the prior biosphere models (global annual budgets listed in Table 1). This study used the year-specific fossil fuel
and cement production inventory from the Open-source Data Inventory for Anthropogenic $CO_2$ (ODIAC-2016) developed by Oda
et al. (2018). Following the approach of Nassar et al. (2013), the monthly ODIAC-2016 inventory is converted from the native
temporal variability into diurnal (hourly) and weekday/weekend variability (courtesy: Sourish Basu and the OCO-2 Science Team).
Wild fire emissions and fuel wood burning emissions were taken from the 3-hourly varying Global Fire Emissions Database
(GFED3) database. Shipping emissions are from the International Comprehensive Ocean-Atmosphere Data Set (ICOADS)
(Corbett and Koehler, 2003; 2004) and aviation emissions are from Aviation Emissions Inventory Code (AEIC) inventory (Olsen
et al., 2013). We used 3D chemical production of $CO_2$ from the oxidation of carbon monoxide, methane and non-methane volatile
organic compounds (Nassar et al., 2010). The shipping, aviation and 3D chemical source are climatological and are taken from the
Bowman (2017) dataset. To simulate oceanic $CO_2$ fluxes, we apply the year-specific 3-hourly posterior estimates from the
CarbonTracker 2016 (CT2016) model constrained with in situ data (Peters et al., 2007; http://carbontracker.noaa.gov). All emission
inventories, besides NEE fluxes, are kept constant between the different inverse model simulations.
**2.4 Observing System Simulation Experiments (OSSEs)**
**2.4.1 OSSE framework**
This study conducted several OSSEs to assess the impact of prior biospheric $CO_2$ models and associated prior uncertainty
specifications on posterior estimates of NEE when assimilating OCO-2 data. To assess the impact of prior fluxes, we conduct four
baseline OSSEs (using the four prior biosphere models) assimilating synthetic land nadir (LN) and land glint (LG) observations
together, plus an additional four OSSEs using just ocean glint (OG) observations. These OSSE simulations were designed in such
a way that the differences in posterior NEE estimates are due solely to the choice of prior biospheric flux (e.g., identical initial
atmospheric $CO_2$ conditions, non-NEE fluxes, OCO-2 sampling frequency, observational data uncertainty, etc.). Furthermore, to
assess the impact of prior uncertainty specifications, we conduct two additional OSSEs (in addition to our baseline prior uncertainty
assumption described in Sect. 2.4.5) with synthetic LN and LG observations using prior error set uniformly to 10% and 100% of
a particular prior NEE flux (CASA-GFED). Table 2 shows the summary of OSSEs conducted for this study. During all OSSE
simulations NEE and oceanic $CO_2$ fluxes are optimized, with all other sources kept constant, in order to be consistent with the
methods commonly used by inverse modeling systems focused on estimating NEE. We optimize oceanic fluxes together with NEE,
although the same ocean fluxes were used for the "truth" and the prior in all OSSE simulations (Sect. 2.3) for simplicity and
because the terrestrial NEE fluxes are the focus of this work. It is noteworthy that the assimilation of land or ocean data in fact do
not produce substantial deviations from the "truth" over the TransCom-3 oceanic regions (Fig. S1). For all OSSE simulations, an
assimilation window of 18 months covering the period from August 1, 2014 to January 31, 2016 was applied. NEE/oceanic fluxes
are optimized for every month of the assimilation window at each surface grid box in the GEOS-Chem model. The analysis of
prior and posterior NEE fluxes is for all months in 2015, treating the other months as spin-up and spin-down periods.



**1**  **2.4.2 Initial CO₂ concentrations**

**2**  We use identical initial atmospheric concentrations of $CO_2$ at August 1, 2014 for: 1) the GEOS-Chem forward model simulations

**3**  generating synthetic $XCO_2$ using the "true" NEE fluxes (Sect. 2.4.3) and 2) for all the OSSEs using variable prior biosphere model

**4**  predictions. The initial $CO_2$ concentrations were generated by running the GEOS-Chem forward model for two years using the

**5**  "true" MsTMIP NEE and other non-NEE $CO_2$ sources. The restart file used for this two year forward model run was taken from

**6**  an earlier GEOS-Chem model simulation constrained with in situ observations (personal communication from Ray Nassar) in order

**7**  to represent a realistic initial condition.

**8**  **2.4.3 Synthetic OCO-2 retrievals**

**9**  In this study, we used synthetic satellite data that is directly representative of version 8 of the OCO-2 product. The OCO-2 satellite

**10**  sensor is in sun-synchronous polar orbit with a repeat cycle of 16 days and a local over-pass time in the early afternoon (Crisp et

**11**  al., 2017). OCO-2 has three different viewing modes: soundings over land from LN and LG and over oceans from OG. The

**12**  algorithm of O'Dell et al. (2012) is used to retrieve column-average dry air mole fraction of $CO_2$ ($XCO_2$) and other retrieval

**13**  variables. OCO-2 $XCO_2$ is retrieved using the following equation using a prior $CO_2$ vertical profile ($\mathbf{c}_a$) and prior $CO_2$ column

**14**  ($XCO_{2\,(a)}$) value,

**15**  $$XCO_2 = XCO_{2\,(a)} + \mathbf{a}^{\mathrm{T}}(\mathbf{c} - \mathbf{c}_a) \qquad (1)$$

**16**  where $\mathbf{c}$ is the true profile of $CO_2$ concentrations, $\mathbf{a}$ is the averaging kernel vector. The individual soundings of OCO-2 are at a

**17**  fine-resolution (24 spectra per second with $< 3$ km$^2$ spatial resolution per sounding), leading to a very large data volume (Crisp et

**18**  al., 2017). This level of detail is lost when the measurements are used in global inverse models with much coarser spatial resolution,

**19**  with numerous individual OCO-2 soundings occur in a single model grid cell. In addition, each sounding does not really provide

**20**  an independent piece of information to the inversion system due to spatial and temporal error correlations. Therefore, we use 10-

**21**  sec averages of the individual $XCO_2$ soundings similar to those developed/described in Basu et al. (2018), however, from an

**22**  updated version with file name: OCO2_b80_10sec_WL04_GOOD_v2.nc. This 10-sec data contains averages of retrievals with a

**23**  "GOOD" quality flag and Warn Level from 1 to 4. Note that in this study we do not use actual retrieved 10-sec $XCO_2$ values for

**24**  our OSSEs, however, synthetic $XCO_2$ data were generated corresponding to the spatio-temporal sampling frequency of version 8

**25**  OCO-2 10-sec data. The synthetic 10-sec $XCO_2$ data is calculated with the $CO_2$ concentration profile simulated using the GEOS-

**26**  Chem forward model (using MsTMIP as the NEE flux model). In this manner we produced synthetic $XCO_2$ data that is

**27**  representative of the "true" atmosphere corresponding to the "true" NEE fluxes used in this study. We archive these synthetic

**28**  $XCO_2$ data and applied them for all OSSEs conducted.

**29**  **2.4.4 Inverse modeling approach**

**30**  The transport of atmospheric $CO_2$ is simulated using GEOS-Chem along with prescribed surface fluxes as input data (see Table

**31**  1). Subsequently, the GEOS-Chem 4D-Var inverse modeling system assimilates synthetic OCO-2 $XCO_2$ data to estimate the

**32**  posterior/optimized monthly-mean NEE and oceanic fluxes at each surface grid of the model. The GEOS-Chem adjoint system

**33**  applies the L-BFGS numerical optimization algorithm with a no-bound option (Liu and Nocedal, 1989). Posterior monthly-

**34**  averaged NEE/oceanic fluxes ($\mathbf{x}$) are inferred for each surface model grid by optimizing a vector of scaling factors $\sigma_j$ for the $j$th

**35**  model grid,

**36**  $$x_j = \sigma_j\, x_{a,j} \qquad (2)$$



where $\mathbf{x}_a$ represents the monthly-mean prior NEE/oceanic fluxes. Scaling factors are assumed to be unity at the first iteration (that
is, prior NEE flux itself is used). The inversion system, as described below, optimizes the scaling factor applied to the monthly-
mean fluxes and the posterior scaling factors are then used to scale prior fluxes to infer posterior $CO_2$ fluxes.
For each iteration, the inversion system uses the forward model simulated profiles of $CO_2$ concentrations mapped to OCO-
2 retrieval levels ($\mathbf{f}(\mathbf{x})$) in each model grid in order to compare with the synthetic OCO-2 observations ($\mathbf{y}$). The observation operator
(M) represents the model simulated $XCO_2$ corresponding to each synthetic OCO-2 retrieval,
$\qquad M = M_a + \mathbf{a}^T (\mathbf{f}(\mathbf{x}) - \mathbf{y}_a)$         (3)
where $M_a$ is the prior $XCO_2$ used in the retrieval of the OCO-2 $XCO_2$ product and $\mathbf{y}_a$ is the corresponding prior profile of $CO_2$
concentrations assumed in the retrieval. The optimization approach used in this work defines the 4D-Var cost function (J) as,
$\qquad J(\boldsymbol{\sigma}) = \frac{1}{2}\sum_i (M_i - \mathbf{y}_i)^T \mathbf{R}_i^{-1} (M_i - \mathbf{y}_i) + \frac{1}{2}(\boldsymbol{\sigma} - \boldsymbol{\sigma}_a)^T \mathbf{P}^{-1} (\boldsymbol{\sigma} - \boldsymbol{\sigma}_a)$     (4)
where $\mathbf{y}_i$ is the vector of synthetic OCO-2 $XCO_2$ data across the assimilation window, with 'i' being the number of $XCO_2$ data.
Furthermore, $\mathbf{R}$ and $\mathbf{P}$ are the observational error covariance matrix and prior error covariance matrix, respectively.
**2.4.5 Prior flux uncertainty**
For a perfect optimization, the prior error covariance matrix ($\mathbf{P}$) assumed in the inversion should equal the uncertainty of the prior
model used. However, the estimation of prior error statistics is a challenging task due to the lack of flux evaluation data. Previous
studies have used a range of techniques to characterize prior error covariance: by developing the full error covariance matrix with
assumed error correlations (Basu et al., 2013), based on a fraction of heterotrophic respiration (Basu et al., 2018), conducting
Monte Carlo simulations (Liu et al., 2014), using standard deviations and absolute differences between several different prior flux
models (Baker et al., 2006a; 2010) and applying globally uniform prior flux uncertainty values to satisfy the posterior
$\chi^2$(normalized cost function) = 1 criteria (Deng et al., 2014). During this study, a 1σ standard deviation (SD) of the four prior
biosphere models (see Sect. 3.1 for the description of SD values) is considered to be the measure of uncertainty in the prior
knowledge of "bottom-up" model predicted biospheric $CO_2$ fluxes. The SD of the four prior NEE estimates is applied in the prior
error covariance matrix and no spatial or temporal correlations are taken in to account. This assumption is reasonable as optimized
fluxes are at coarse spatio-temporal scales (monthly mean fluxes at horizontal resolutions of > 400 $km^2$) and is representative of
the majority of inverse modeling studies assimilating $CO_2$ satellite data (e.g., Baker et al., 2010; Liu et al., 2014; Deng et al., 2014).
Finally, since the inverse modeling system applied in this work optimizes scaling factors, we use the square of the fractional prior
error in the $\mathbf{P}$ matrix, where the fractional error is calculated for each individual prior model as the SD of the four prior models
divided by the absolute value of the NEE magnitude. For generating prior error for the oceanic fluxes, we follow the same method
we adopted for generating prior errors in NEE. The SD of four different state-of-the-science oceanic $CO_2$ flux datasets (NASA-
CMS $CO_2$ oceanic flux from Bowman (2017), CarbonTracker 2016 prior ocean data (CT2016; http://carbontracker.noaa.gov),
Takahashi et al. (2009) and Landschützer et al. (2016; 2017)) was calculated to generate prior error values.
**2.4.6 $XCO_2$ uncertainty**
As described in Sect. 2.4.3, the synthetic $XCO_2$ used in this study is calculated at the spatio-temporal sampling frequency of OCO-
2 10-sec average dataset. Although we use synthetic $XCO_2$, we apply the same observation error statistics generated with the actual
OCO-2 $XCO_2$ 10-sec dataset in order to develop the observational error covariance matrix ($\mathbf{R}$). The final observation error for the
10-sec average data is generated as a quadratic sum of the retrieval error from individual OCO-2 soundings, 10-sec averaging error,
and a 'model representation error' as described in Basu et al. (2018). Similar to prior error statistics, we neglect observation error
correlations and assume a diagonal observational error covariance. No random perturbations were added to the synthetic $XCO_2$



used in this study, as the goal of this work was not to quantify the analytical posterior flux uncertainty but instead was aimed to analyze the spread among posterior NEE estimates. We note that other inverse modeling groups assimilating OCO-2 data also do not add random perturbations to the data and use the same error statistics generated along with the OCO-2 10-sec product that are applied in this study (e.g., Basu et al., 2018; Crowell et al., in prep). During this study the square of the observation error for the 10-sec average data was applied as the diagonal in the **R** matrix. From our initial OSSE tests it was determined that the use of 10-sec error statistics led to posterior $\chi^2$ (normalized cost function) values that were much lower than unity. Therefore, we divided the 10-sec error values uniformly by a factor of 5 to approximately satisfy the $\chi^2 = 1$ criteria for all the OSSEs. This deflation procedure reduced the average 10-sec observational error values for LN+LG (OG) data from ~1.5 (~0.9) ppm to ~0.3 (~0.2) ppm. These procedures give more confidence to the observational data and lead to results in this study which can be assumed as the lower limit of the impact of prior model flux and uncertainty statistics in inverse model estimates.

### 2.4.7 Evaluation of OSSE results

During this study, the posterior NEE values from the OSSEs are compared to the "true" fluxes to assess accuracy and also inter-compared to assess the spread in posterior estimates due to the assumed prior NEE and prior error statistics. The primary statistical parameters used to evaluate the spread in posterior NEE fluxes are the SD (hereafter the term "spread" will be used to represent SD) and range (difference between maximum and minimum in NEE). The SD/spread and range of posterior NEE estimates, when using the different prior models, will provide an understanding of the spatio-temporal residual impact of the prior models in "top down" estimates of global/regional NEE fluxes when assimilating OCO-2 data.

In order to evaluate the spatio-temporal variability of prior and posterior regional NEE fluxes, we aggregate individual model grids to The Atmospheric Tracer Transport Model Intercomparison Project-3 (TransCom-3) land regions (TransCom-3 regions illustrated in Fig. S2). To further interpret the OSSE results, we produce additional classifications of three broad hemisphere-scale TransCom-3 land regions: Northern Land (NL), Tropical Land (TL) and Southern Land (SL). TL includes Tropical South America, North Africa and Tropical Asia; SL includes South American Temperate, South Africa and Australia; and NL includes the other five land regions. The evaluation of SD and range of prior model and posterior/optimized NEE fluxes were calculated for the 11 individual TransCom-3 regions, for the three hemisphere-scale TransCom-3 regions and globally. Throughout the manuscript, seasonally-averaged prior and posterior NEE fluxes will be discussed and these seasons are presented with respect to the Northern Hemisphere.

### 2.4.8 Pseudo data assimilations

In order to test our OSSE framework, we first run four "pseudo" experiments by conducting inverse modeling studies using "pseudo" surface observations. These test OSSE simulations were conducted for five-month assimilation windows for two separate seasons (November 2014 to March 2015 (analysis for winter, DJF) and May 2015 to September 2015 (analysis for summer, JJA)) using all four prior model NEE values separately. Simulated hourly concentrations of $CO_2$ for all surface grids of GEOS-Chem are taken as "pseudo" surface observations. In order to check whether the model framework can converge to the "truth", a simple controlled experiment was performed assuming a very small observational data uncertainty (0.001%) and with the prior flux uncertainty set equal to the absolute magnitude of the "truth" - prior NEE (divided by the absolute value of the NEE magnitude for that respective prior model). The robustness of the flux inversions conducted in the subsequent sections is validated by the results of these "pseudo" tests. Figure S3 shows the results of the four "pseudo" tests using the four different NEE flux model predictions as the prior information. From this figure it is apparent that, regardless of the prior NEE assumed, posterior NEEs were able to reproduce the "truth" with near perfect accuracy for all TransCom-3 regions, with the range between the posterior NEEs typically



1. approaching ~0 PgC yr$^{-1}$. This test also demonstrates that a "perfect" assimilation (using uniform and dense surface data coverage,

2. highly accurate data and known/loose prior uncertainty) is almost insensitive to the prior assumed. Having tested the robustness of

3. our inversion setup, we feel confident in presenting the output from our OSSE framework, using synthetic OCO-2 remote-sensing

4. data, in the following sections.

5. **3. Results and discussion**

6. **3.1 Prior NEE fluxes**

7. Figure 1 shows the seasonally-averaged multi-model-mean and SD of the NEE fluxes from the four prior biosphere models used

8. in the OSSE simulations (individual prior model and "true" seasonally-averaged NEE fluxes are displayed in Fig. S4). This figure

9. shows the main features of NEE that are expected, such as the Northern Hemispheric fall/winter maximum in R$_e$ and summer

10. maximum in GPP due to the seasonality of photosynthesis and respiration. Figure 1 also shows the spread of the four prior model

11. fluxes (used as prior uncertainty), which is typically highest over the Temperate regions of the Northern Hemisphere in the spring

12. and summer and high in the Tropical regions during all seasons (the left panel of Figure S5 shows a spatial map of the corresponding

13. range of the four prior model fluxes). Furthermore, Fig. S6 shows the time-series of monthly-mean prior NEE fluxes, and the

14. corresponding prior error uncertainty (error bars), for the individual prior models. From this figure it can be seen that the 1σ prior

15. uncertainties for the global land are ~50-70% of the total NEE for the different prior models, with variability among other

16. TransCom-3 land regions.

17. Table 3 displays the statistics of the prior NEE multi-model-mean and SD and range for the 11 individual TransCom-3

18. land regions. The SD values for prior NEE fluxes range from ~20% to frequently > 100% of the multi-model NEE mean for

19. different regions/seasons with significant NEE fluxes (hereafter this refers to regions/seasons with NEE flux ≥ 0.5 PgC yr$^{-1}$). When

20. comparing the magnitude of NEE between the four prior models, it can be seen from this table that the range in NEE values are

21. large in some regions (up to 6 PgC yr$^{-1}$). In general, all regions/seasons tend to have at least an ~1 PgC yr$^{-1}$ range among the four

22. prior models, indicating the large diversity in NEE predicted by current "bottom-up" biosphere models (Table 3). Figure S6 shows

23. the time-series of monthly-mean NEE for individual prior models averaged over the globe, hemispheric-scale land regions (NL,

24. TL, SL) and the 11 individual TransCom-3 land regions. It can be seen from this figure that the majority of the seasonality in the

25. global NEE flux is controlled by the NL regions. Figure S6 also shows that the spread in prior NEE fluxes in general is larger for

26. TL and SL regions compared to the NL, except for the North American Temperate region. Furthermore, when focusing on

27. individual models, differences in NEE seasonality are evident. The impact of these differences among the four prior biospheric

28. CO$_2$ flux models on simulated XCO$_2$ and posterior estimates of global/regional NEE fluxes is evaluated in the following sections.

29. **3.2 Simulated XCO$_2$**

30. Figure 2 shows the number of observations sampled in the OCO-2 LN and LG modes during the different seasons of 2015 summed

31. in each model grid. Large spatio-temporal variability can be seen in the OCO-2 observation density, with the largest values over

32. regions with minimal cloud coverage (e.g., desert regions of North/South Africa, Middle East, Australia, etc.). The opposite is true

33. for many Tropical regions (e.g., Amazon, central Africa, Tropical Asia, etc.) where cloud occurrence is prominent and the number

34. of OCO-2 observations is lowest. From Fig. 2 it can also be seen that the OCO-2 observation density has noticeable seasonality.

35. For example, during the winter months low numbers of OCO-2 observations are made in the Northern Boreal regions and the

36. largest amounts are observed during the summer. Furthermore, larger numbers of OCO-2 observations are made in the SL during

37. the summer (JJA) compared to other seasons.



1. The seasonally-averaged multi-model-mean GEOS-Chem simulated $XCO_2$ using the four prior model NEE fluxes is
2. shown in the right panel of Fig. 2. The most notable feature in this figure is the Northern Hemisphere seasonality, with higher
3. $XCO_2$ concentrations in the winter months and lowest $XCO_2$ values in the growing seasons of the summer. Seasonality in model-
4. predicted $XCO_2$ values is also evident in the TL and SL, with largest values in the autumn and lowest values in the spring. Figure
5. S7 (left panel) shows the range of $XCO_2$ values simulated using the four prior model NEE fluxes. The differences between
6. individual model simulations of $XCO_2$ values deviated among themselves by up to ~10 ppm. These large differences in $XCO_2$
7. values across the four different prior NEE flux models show that the choice of prior NEE has a large impact on simulated $XCO_2$
8. values.

9. **3.3 Optimized global NEE fluxes**

10. From Table 1 it can be seen that annual global mean posterior NEE flux, when using the different prior models and assimilating
11. synthetic LN+LG OCO-2 $XCO_2$, ranges from -4.11 to -4.36 PgC yr$^{-1}$, which are generally close in magnitude to the "true" flux of
12. -4.31 PgC yr$^{-1}$. Although these posterior NEEs generally converged to the "truth", there are some remaining differences, with an
13. annual global mean posterior NEE range of 0.25 PgC yr$^{-1}$ (~6% of the multi-model-mean posterior NEE; Table 1). From the results
14. of the OSSE simulations, it was found that the spread and range in $XCO_2$ simulated using the optimized posterior NEE fluxes was
15. greatly reduced compared to the spread in $XCO_2$ simulated using prior NEE fluxes. This is evident from the right panel of Fig. S7
16. where $XCO_2$ simulated using posterior NEE fluxes differ among themselves by < 0.5 ppm, which is greater than an order of
17. magnitude lower, on average, than the spread among $XCO_2$ simulated using prior NEEs.

18. Figure S5 shows the spatial distribution of the range of prior and posterior NEEs. As expected, the range in optimized
19. posterior NEE flux estimates starting from the four separate prior models was substantially reduced compared to the spread in prior
20. NEE fluxes. However, the posterior NEE fluxes for individual surface grid boxes of the model still depict some residual range
21. among the posteriors, with the largest residuals being found across South America and South Africa in all seasons and in Temperate
22. regions of the Northern Hemisphere in the spring months. As shown in Fig. S5, the geographical pattern of the range of prior and
23. posterior NEEs does not indicate any noticeable correlations. From comparing Figs. S5 and S7, it is apparent that the spread in
24. posterior $XCO_2$ is significantly reduced in all regions of the globe compared to prior model simulations, however, while posterior
25. NEE values are reduced compared to the prior, noticeable residual spread remains in some regions. This emphasizes the fact that
26. the OSSEs successfully converge to match the synthetic OCO-2 $XCO_2$ values by optimizing NEE in different ways depending on
27. the prior NEE model used. The following sections investigate the regional differences in posterior NEE estimates due to the residual
28. impact of prior biospheric $CO_2$ flux predictions.

29. **3.4 Optimized regional NEE fluxes**

30. Figure 3 shows the seasonally-averaged "true", prior and posterior NEE flux values for the 11 individual TransCom-3 land regions
31. (with detailed statistics in Table 3 and monthly-mean time-series in Fig. S8). The first thing noticed from this figure is that all
32. posterior NEE values, using variable priors, tend to reproduce the "truth" in most TransCom-3 land regions. From Fig. 3 it can
33. also be seen that the assimilation of synthetic OCO-2 LN+LG $XCO_2$ retrievals resulted in a large reduction in the range among the
34. four modeled NEE values (Table 3 shows the corresponding SD values). The reduction in the SD of NEE in most regions/seasons,
35. calculated as $100 \times (1 - $ (posterior NEE SD)/(prior NEE SD)) is generally > 70% and up to 98%. However, the range of seasonal
36. mean posterior NEEs over individual TransCom-3 regions is still as large as 1.4 PgC yr$^{-1}$ when applying different prior NEE, with
37. the largest ranges occurring in Northern Boreal regions (North America Boreal, Eurasian Boreal and Europe) in winter months.
38. During the spring and summer months, regions in the TL (e.g., Tropical Asia) and SL (e.g., South American Temperate, South

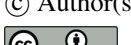



Africa) have ranges in posterior NEEs up to ~0.5 PgC yr$^{-1}$. The larger residual range among posterior NEE estimates for winter
months in Northern Boreal regions is likely due to the insufficient OCO-2 observations during this time (see Fig. 2), while the
larger range in the TL and SL regions is due to differences in the priors (see Fig. 1). This demonstrates that the impact from the
prior model has regional and seasonal variability depending on: 1) the spatio-temporal flux variabilities inherent in prior NEEs and
2) the observation density and coverage of synthetic OCO-2 data. Figure 3 and Table 3 show that the seasonally-averaged posterior
NEE spread varies from ~10% to ~50% of the multi-model-mean for different TransCom-3 land regions with significant NEE
fluxes. When evaluating this residual spread in posterior NEEs on a global average, seasonally-averaged values ranged from ~10%
(JJA) to ~20% (DJF) of the posterior NEE mean. These statistics reveal that the impact of prior models lead to a much larger
spread/range for regional/seasonal posterior fluxes (up to ~50%) compared to annual global averaged values (6%). This emphasizes
that while OCO-2 observations on average constrain global, hemispheric and regional biospheric fluxes, noticeable residual
differences in posterior NEE flux estimates remain due to the choice of prior model values. Overall, the results of this evaluation
suggest that when inter-comparing inverse model results assimilating similar OCO-2 observational data, differences in posterior
NEE in regions with significant NEE fluxes could vary by up to ~50% when using different prior flux assumptions.
**3.5 Impact of prior uncertainty**
Results of this study thus far have demonstrated the sensitivity of posterior NEE estimates to prior NEE flux assumptions. In this
section, the sensitivity of posterior NEE estimates to the assumed prior uncertainty is tested, when assimilating synthetic OCO-2
LN+LG XCO$_2$ observations. The general importance of prior uncertainty values is highlighted in the TL regions. In these regions
the largest differences in prior models are calculated, thus largest prior uncertainty is assigned, resulting in larger deviations from
the prior and posterior NEE spread similar to other TransCom-3 land regions (see Fig. 3). In order to quantify the sensitivity of
posterior NEE to prior uncertainty statistics, a single prior NEE flux model (CASA-GFED) is applied in the OSSE framework,
with variable prior flux uncertainty assumptions. Two additional OSSE simulations (in addition to the baseline simulations using
the SD of the four prior models as the prior uncertainty; see right panel of Fig. 1 for SD maps) are performed using prior NEE
magnitudes from CASA-GFED and setting the prior uncertainty uniformly as 10% and 100% of the CASA-GFED NEE values.
Figure 4 shows the results of these additional OSSE simulations over the TransCom-3 land regions. From this figure it can be seen
that the range of seasonal mean posterior NEEs over individual TransCom-3 regions vary from ~0.1 to > 1 PgC yr$^{-1}$ when applying
variable prior error assumptions. Seasonally-averaged posterior NEE SD varies from ~10-50% of the multi-model-mean for
different TransCom-3 land regions with significant NEE fluxes. On a global average, seasonal-average SD values range from
~15% (JJA) to ~30% (DJF) of the posterior NEE mean. Note that these posterior NEE SD/range values here are similar to the
baseline OSSEs conducted by changing prior NEE flux magnitudes (see Fig. 3). However, when comparing Fig. 4 and 3 (and Fig.
S9), it is noticed that posterior NEE estimates are more sensitive to prior error assumptions compared to prior flux values in some
seasons/regions of TL and SL (e.g., Northern Africa, Southern Africa, South America Temperate). It appears that NEE estimates
during this study are more sensitive to prior error assumptions when sufficient observations are available and large differences
between the prior and "truth" are present. Also, from Fig. 4 it can be seen that prior uncertainty assumptions in the baseline runs
(using SD of prior models) and the assumption of 100% prior uncertainty tend to reproduce the "truth" more accurately than NEE
estimates using 10% prior error. Overall, the results demonstrate that the posterior NEE fluxes over TransCom-3 land regions are
in general similarly sensitive (up to ~50%) to the specification of prior flux uncertainties and the choice of bottom-up prior
biospheric NEE model estimates.



**3.6 OCO-2 ocean data**
This portion of the study investigates the impact of assimilating OCO-2 OG $XCO_2$ data on posterior NEE flux estimates in our
OSSE framework. To do this, four additional OSSE simulations were conducted with the four prior model NEEs when only
assimilating synthetic OG retrievals (instead of LN+LG) in the inversions (everything else remains the same as in the baseline
simulations). Figure 5 shows the results of these four additional OSSE simulations averaged over the TransCom-3 land regions.
From these simulations it can be seen that OCO-2 OG indeed reduces the range in posterior NEE flux estimates, when applying
different priors, compared to prior model predictions, and can generally reproduce the "truth". On average, the spread in posterior
NEE fluxes is ~20% to ~50% of the multi-model-mean for different TransCom-3 land regions with significant NEE fluxes. As
expected, the comparison of Fig. 3 and 5 suggests that LN+LG data is better able to constrain biospheric $CO_2$ fluxes compared to
OG data, as the spread among the posteriors is generally lower in LN+LG only assimilations (~70% lower on a global average)
compared to OG data only assimilations. However, there were some cases where OSSE simulations using OCO-2 OG data alone
did in fact result in slightly lower posterior NEE spreads in some TransCom-3 land regions compared to LN+LG assimilations
runs (e.g., Northern Boreal regions during summer months and Australia during winter months). Overall, our OSSE simulations
using the OCO-2 OG data demonstrate the importance of these oceanic retrievals to constrain land NEE fluxes, as the posterior
NEE range is much lower compared to prior NEE estimates (see Fig. 5). This generally agrees with previous studies that
demonstrated the importance of satellite data over the ocean in constraining NEE fluxes over land regions (e.g., Deng et al., 2016).
**4. Conclusions**
To the best of our understanding, this is the first study directly quantifying the impact of different prior global land biosphere
models on the estimate of terrestrial $CO_2$ fluxes when assimilating OCO-2 satellite observations. We conducted a series of OSSEs
that assimilated synthetic OCO-2 observations applying four state-of-the-science biospheric $CO_2$ flux models as the prior
information. These controlled experiments were designed to systematically assess the impact of prior NEE fluxes and the impact
of prior error assumptions on "top down" NEE estimates using OCO-2 data. The OSSEs incorporated NEE fluxes from the NASA-
CASA, CASA-GFED, SiB-4 and LPJ biosphere models as prior estimates and variable prior flux error assumptions.

24          We found that the assimilation of synthetic OCO-2 $XCO_2$ retrievals resulted in posterior monthly/seasonal NEE estimates

that generally reproduced the assumed "true" NEE globally and regionally. However, spread in posterior NEE exists in regions
during seasons with poor data coverage, such as the Northern Boreal regions and some of the Tropical and Southern Hemispheric
regions (e.g., South American Temperate, South Africa, Tropical Asia). This spread among posterior NEEs is likely due to the
insufficient OCO-2 observations during winter over Northern Boreal regions and the large range among the priors in some of the
Northern Boreal, Tropical and Southern Hemispheric regions. Residual spread from ~10% to 50% in seasonally-averaged posterior
NEEs in TransCom-3 land regions with significant NEE flux were calculated due to using different prior models in inverse model
simulations. We also found similar spreads in the magnitudes of posterior NEEs by conducting additional OSSEs using a single
prior NEE flux model with variable prior flux uncertainty assumptions. While the spread in posterior NEE estimates, when using
variable prior error statistics, was similar to when applying variable NEE flux models, the impact was larger in some seasons in
the TL and SL regions. We determined that while OCO-2 observations on average constrain global, hemispheric and regional
biospheric fluxes, noticeable residual differences (up to ~20-30% globally and 50% regionally) in posterior NEE flux estimates
remain that were caused by the choice of prior model values and the specification of prior flux uncertainties.

37          There have been previous studies that investigated similar scientific objectives, such as the impact of prior uncertainties

on inverse model estimates of NEE (Gurney et al., 2003; Chevalier et al., 2005; Baker et al., 2006a; 2010). The sensitivity of $CO_2$
flux inversions to the specification of prior flux information was first assessed by Gurney et al. (2003) using ground-based in situ





data. One main conclusion from Gurney et al. (2003) is that $CO_2$ flux estimates were sensitive to the prior flux uncertainty over regions with limited observations and insensitive over data-rich regions. Chevallier et al. (2005) suggested the importance of an accurate formulation of prior flux uncertainty by conducting 4D-Var assimilation of satellite column retrievals of $CO_2$. Baker et al. (2010) investigated the importance of assumed prior flux uncertainties by conducting sensitivity tests that mistuned the assimilations by using incorrect prior flux errors. Finally, Baker et al. (2006a, 2010) suggested the need for realistic prior models in the 4D-Var assimilations using OCO synthetic satellite $CO_2$ data. The results of this research are generally consistent with the findings of these past studies. However, in comparison with these previous efforts, our study is a step forward, because we quantify the specific impact of prior model NEE spatio-temporal magnitude and prior uncertainties in optimizing regional and seasonal NEEs using satellite data in a more controlled manner by applying an OSSE framework.

The results of this study suggest the need to be aware of the residual impact from prior assumptions for $CO_2$ global flux inversions, especially for regions and times 1) where current "bottom-up" biosphere models diverge greatly and 2) without sufficient observational coverage from space-borne platforms. For example, larger spread in posterior NEE estimates were calculated in portions of the Northern Boreal regions that tend to have insufficient satellite data coverage and moderate differences among prior biosphere models. In addition to these Northern Boreal regions, Tropical and Southern Hemispheric regions with large spread among prior biosphere models, which are assigned higher prior uncertainty values resulting in largely reduced spreads in posterior NEE estimates (large deviation from the prior), still have residual impact from the prior NEE predictions regardless of the fact that OCO-2 data is dense in these regions. Results of this study also indicate that in some regions/seasons of the TL and SL, inverse model estimates of NEE can be more sensitive to prior error statistics compared to prior flux values. Overall, in data-poor regions/times, posterior estimates from inversion techniques relying on Bayesian statistics can result in similar estimates to the prior flux, however, with some improvements over broader regions. Additionally, in regions/seasons where uncertainty in NEE fluxes are large (e.g., in the TL where prior model NEE differences are large), inverse model estimates, applying large prior uncertainty values, will still have some residual impact from the choice of prior NEE flux. Finally, care should be given when interpreting flux estimates constrained with real OCO-2 satellite data over some of the regions identified in this study as it is suggested here that residual differences (up to ~20-30% globally and 50% regionally) in posterior NEE flux estimates can be produced by the choice of prior model values and the specification of prior flux uncertainties. In the future, studies should be designed to determine the relative importance of prior flux magnitudes and error assumptions on posterior estimates, in comparison with other error sources in inverse flux estimates (such as transport model errors and observation errors). Finally, the results of this study suggest that multi-inverse model inter-comparison studies should consider the differences in posterior NEE flux estimates caused by variable prior fluxes and error statistics used in different models.

**5. Code and Data Availability**

The forward and inverse model simulations for this work were performed using the GEOS-Chem model which is publicly available at: http://acmg.seas.harvard.edu/geos/. The 10-sec OCO-2 data used to produce synthetic observations during this study are available by request from the OCO-2 Science Team and individual OCO-2 sounding data can be downloaded here: https://oco.jpl.nasa.gov/.

**6. Author Contributions**

SP and MJ designed the methods and experiments presented in the study and analyzed results. CP, VG, DB, KH and BP were instrumental in providing biosphere model and OCO-2 data and guidance when applying these products. DH, JL and DB provided



components implemented in the modeling framework applied during this study. Finally, SP prepared the manuscript with contributions from all listed coauthors.

## Acknowledgements

Sajeev Philip's research was supported by an appointment to the NASA Postdoctoral Program at the NASA Ames Research Center, administered by Universities Space Research Association under contract with NASA. DKH recognizes support from NA14OAR4310136. Resources supporting this work were provided by the NASA High-End Computing Program through the NASA Advanced Supercomputing Division at NASA Ames Research Center. We thank the OCO-2 Science Team for providing the version 8 OCO-2 product. We also thank the OCO-2 Flux Inversion Team, GEOS-Chem model developers, CASA-GFED team and NASA Carbon Monitoring System program for the free availability of their products. CarbonTracker CT2016 prior and posterior ocean fluxes were provided by National Oceanographic and Atmospheric Administration's Earth System Research Laboratory, Boulder, Colorado, USA from the website at http://carbontracker.noaa.gov. We are thankful to Sourish Basu, Feng Deng, Ray Nassar and Tom Oda for sharing data. We thank the support from the Earth Science Division of NASA Ames Research Center. The views, opinions and findings contained in this report are those of the authors and should not be construed as an official NASA or United States Government position, policy, or decision.

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





**Table 1: Prior and (Posterior) global annual mean NEE fluxes and CO₂ emission inventories (PgC yr⁻¹) for the year 2015 used in the**
**OSSE simulations during this study.**

| NEE Model | NEE flux (PgC yr⁻¹) |
|---|---|
| MsTMIP[1,2] | -4.31 |
| NASA-CASA[3] | -1.86 (-4.14) |
| CASA-GFED[4] | -2.42 (-4.24) |
| SiB-4[5] | 0.95 (-4.11) |
| LPJ[6] | -5.53 (-4.36) |
| **Inventory** | **CO₂ emission (PgC yr⁻¹)** |
| Fossil fuel[7] | 9.86 |
| Ocean[8] | -2.41 |
| Biomass burning[9] | 2.05 |
| Fuel wood burning[9] | 0.50 |

[1]MsTMIP NEE dataset is representative of the year 2010 and is an ensemble mean of 15 different NEE models.
[2]Huntzinger et al., 2013; 2016; Fisher et al., 2016a,2016b
[3]Potter et al., 1993; 2012a; 2012b
[4]Potter et al., 1993; Randerson et al., 1996
[5]Haynes et al., 2013; Baker et al., 2013
[6]Sitch et al., 2003; Poulter et al., 2014
[7]Oda et al., 2018; Nassar et al., 2013
[8]CarbonTracker CT2016; Peters et al., 2007
[9]CASA-GFED3; van der Werf et al., 2004; 2006; 2010



1 **Table 2: Summary of the different OSSEs conducted during this work.**

| Experiment (# of OSSEs) | OCO-2 XCO$_2$ Mode | Prior NEE Model | NEE Uncertainty |
|---|---|---|---|
| Variable Prior NEE (4) | LN + LG | All[1] | Multi-Model SD[2] |
| Variable Prior NEE (4) | OG | All[1] | Multi-Model SD[2] |
| Variable Prior Uncert. (2) | LN + LG | CASA-GFED | Uniform 10%/100% |

2 **[1] NASA-CASA, CASA-GFED, SiB-4 and LPJ**

3 **[2] SD = standard deviation**





**Table 3: Seasonally-averaged NEE (PgC yr⁻¹) averaged over the 11 TransCom-3 land regions (refer to Fig. S2) for the MsTMIP ("truth"),**
**multi-model prior mean and multi-model posterior mean (PgC yr⁻¹). The differences between the prior and posterior model NEE values**
**are presented as SD (1σ) and range. Prior model values are presented in standard font and posterior estimates are in bold. Seasons are**
**represented as Winter (W): December-February, Spring (Sp): March-May, Summer (Su): June-August and Fall (F): September-**
**November.**

| Region* | NEE: Truth W | Sp | Su | F | NEE: Mean W | Sp | Su | F | NEE: Standard Deviation W | Sp | Su | F | NEE: Range W | Sp | Su | F |
|---|---|---|---|---|---|---|---|---|---|---|---|---|---|---|---|---|
| **1** | 1.1 | -0.2 | -2.8 | 0.7 | 1.1 **1.0** | 0.4 **0.3** | -3.4 **-2.7** | 1.3 **0.9** | 0.5 **0.4** | 0.5 **0.1** | 0.8 **0.2** | 0.5 **0.1** | 1.2 **0.9** | 1.0 **0.3** | 1.9 **0.5** | 1.2 **0.2** |
| **2** | 1.5 | -1.5 | -2.3 | 0.0 | 1.9 **1.6** | -1.0 **-1.6** | -3.2 **-2.1** | 0.6 **-0.2** | 1.1 **0.3** | 1.3 **0.1** | 2.0 **0.2** | 0.8 **0.1** | 2.3 **0.6** | 3.2 **0.2** | 4.3 **0.4** | 1.6 **0.2** |
| **3** | -0.6 | -0.2 | -1.4 | -1.3 | 0.4 **-0.8** | 0.6 **0.2** | -0.7 **-1.2** | 0.1 **-1.3** | 0.1 **0.1** | 0.8 **0.2** | 1.4 **0.0** | 2.6 **0.1** | 0.2 **0.1** | 1.8 **0.3** | 3.1 **0.1** | 5.9 **0.2** |
| **4** | -1.5 | -0.4 | 0.7 | -0.4 | -1.1 **-1.3** | -0.1 **-0.4** | 0.6 **0.7** | 0.0 **-0.1** | 0.9 **0.1** | 0.4 **0.2** | 0.8 **0.1** | 0.9 **0.1** | 2.0 **0.1** | 0.9 **0.5** | 1.9 **0.2** | 2.0 **0.3** |
| **5** | 1.3 | 1.0 | -1.9 | -2.0 | 0.7 **0.9** | 0.6 **0.4** | -0.8 **-1.7** | -1.4 **-2.0** | 1.0 **0.2** | 0.8 **0.1** | 0.9 **0.1** | 1.2 **0.2** | 2.3 **0.5** | 1.6 **0.3** | 2.0 **0.2** | 3.0 **0.5** |
| **6** | -2.4 | -1.6 | 1.2 | 1.2 | -0.9 **-1.9** | -1.1 **-1.8** | 0.3 **0.8** | 0.7 **1.2** | 0.5 **0.2** | 1.0 **0.1** | 1.2 **0.2** | 0.7 **0.1** | 1.2 **0.3** | 2.2 **0.2** | 2.6 **0.5** | 1.7 **0.2** |
| **7** | 1.2 | 1.0 | -4.6 | 1.2 | 1.5 **1.2** | 0.9 **0.9** | -5.6 **-4.8** | 2.0 **1.3** | 0.9 **0.7** | 1.0 **0.1** | 1.6 **0.1** | 0.6 **0.2** | 1.8 **1.4** | 2.2 **0.3** | 3.3 **0.2** | 1.3 **0.6** |
| **8** | 1.4 | 0.0 | -1.9 | -0.4 | 1.4 **1.3** | -1.0 **-0.2** | -2.3 **-2.3** | 0.1 **-0.7** | 1.3 **0.3** | 0.9 **0.1** | 2.5 **0.2** | 1.7 **0.1** | 3.0 **0.6** | 2.1 **0.3** | 5.7 **0.5** | 3.7 **0.2** |
| **9** | -0.1 | 0.4 | -0.3 | -0.5 | 0.1 **0.3** | 0.0 **0.6** | -0.4 **0.1** | -0.5 **-0.4** | 0.4 **0.2** | 0.4 **0.1** | 0.4 **0.3** | 0.8 **0.0** | 0.8 **0.5** | 0.7 **0.2** | 0.8 **0.5** | 1.9 **0.1** |
| **10** | -0.6 | -0.6 | -0.2 | -0.7 | -0.1 **-0.6** | 0.0 **-0.7** | 0.1 **-0.4** | -0.1 **-0.8** | 0.4 **0.1** | 0.3 **0.1** | 0.3 **0.1** | 0.6 **0.1** | 0.9 **0.2** | 0.6 **0.2** | 0.7 **0.3** | 1.2 **0.3** |
| **11** | 2.8 | -1.7 | -3.2 | 1.6 | 2.3 **2.6** | -0.9 **-1.2** | -3.7 **-3.1** | 1.6 **1.3** | 0.4 **0.6** | 1.2 **0.2** | 1.5 **0.1** | 0.6 **0.1** | 0.9 **1.3** | 2.5 **0.5** | 3.5 **0.3** | 1.5 **0.3** |

**\*TransCom-3 region name and location displayed in Fig. S2.**





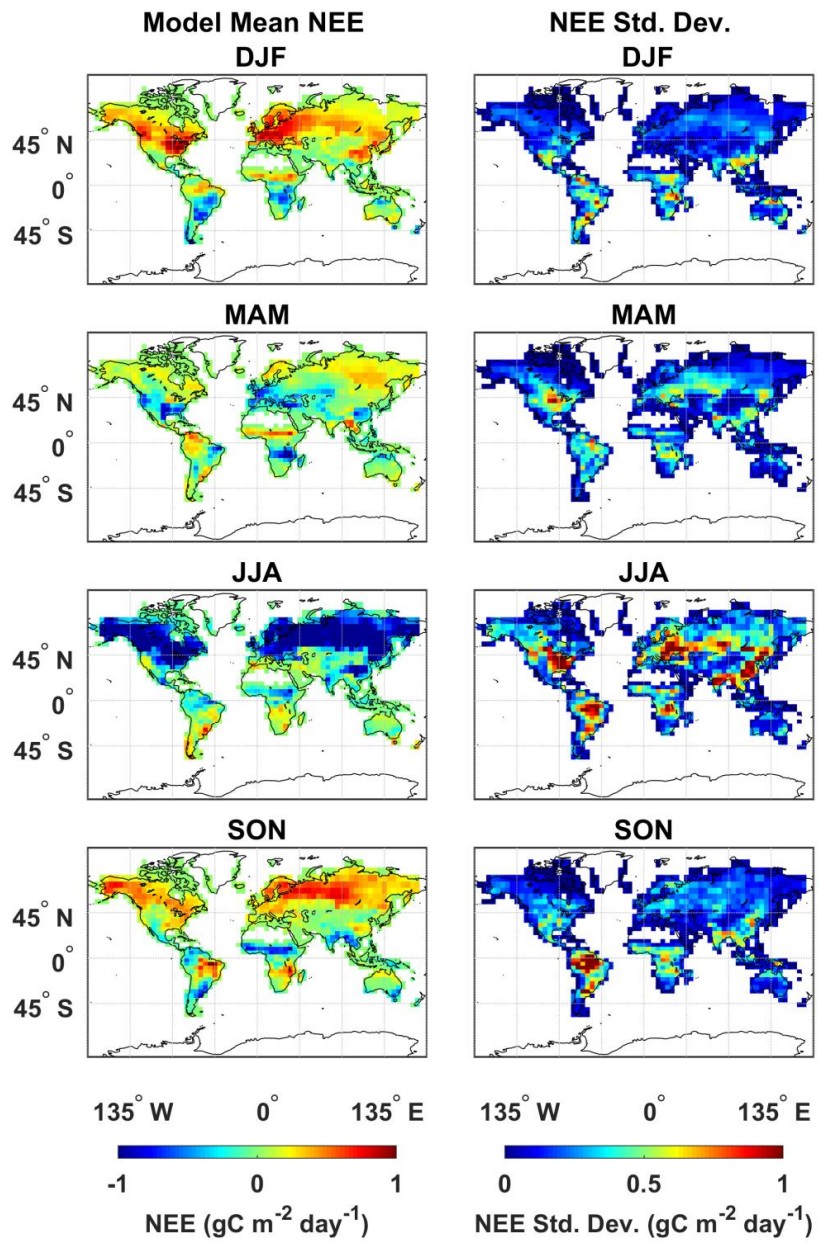

**Figure 1: Prior multi-model (NASA-CASA, CASA-GFED, SiB-4 and LPJ biosphere models) seasonally-averaged NEE (gC m$^{-2}$ day$^{-1}$)**
**(left column) and NEE standard deviation (gC m$^{-2}$ day$^{-1}$) (right column) for the year 2015.**





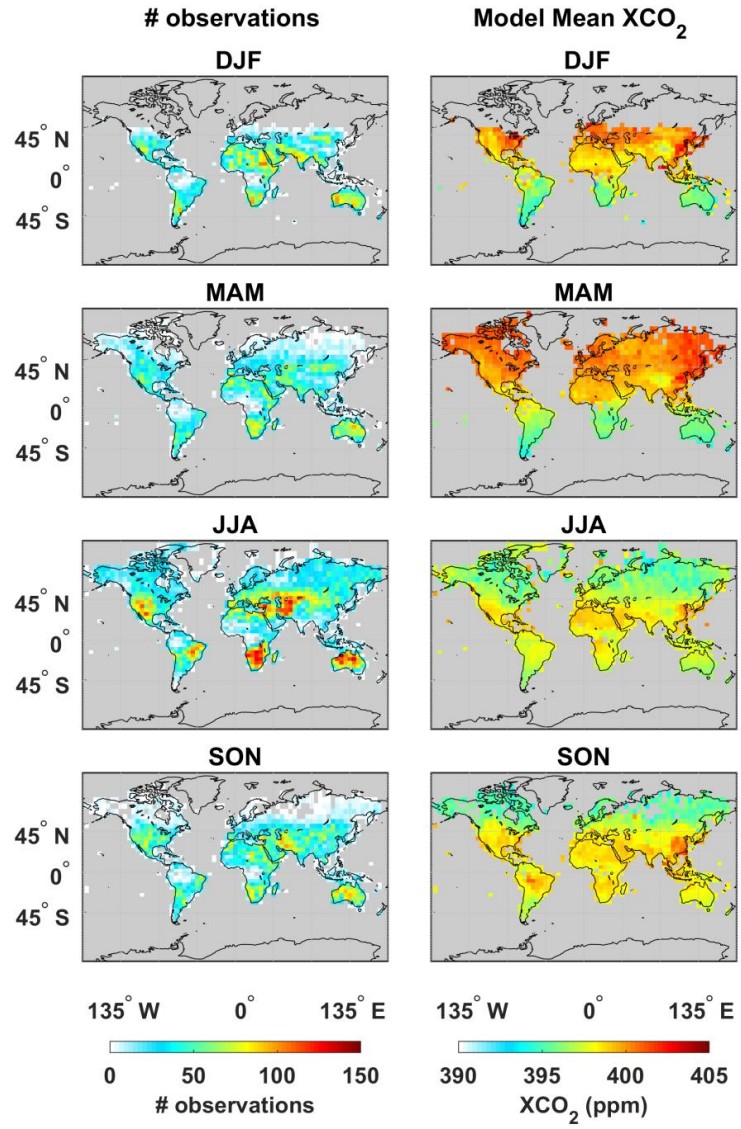

**Figure 2: Total number of OCO-2 LN and LG XCO₂ observations (left column) and the corresponding seasonally-averaged multi-model**
**(NASA-CASA, CASA-GFED, SiB-4 and LPJ biosphere models) mean GEOS-Chem-simulated prior XCO₂ (ppm) (right column) in 2015.**





**Figure 3: Seasonally-averaged NEE (PgC yr⁻¹) averaged over the 11 TransCom-3 land regions from MsTMIP ("truth") versus the prior**
**biosphere models (NASA-CASA, CASA-GFED, SiB-4 and LPJ) (left column), posterior estimates (middle column) from the OSSE**
**simulations and the corresponding range of prior and posterior NEE estimates (right column). The synthetic observations in these OSSE**
**simulations correspond to the OCO-2 LN+LG observing modes.**



**Figure 4: Seasonally-averaged NEE (PgC yr⁻¹) averaged over the 11 TransCom-3 land regions from MsTMIP ("truth") versus CASA-GFED prior biosphere model (left column), posterior estimates with the three different prior uncertainties (middle column) and the corresponding range of posterior NEE (right column). The synthetic observations in OSSE simulations correspond to the OCO-2 LN+LG observing modes.**





**Figure 5: Seasonally-averaged NEE (PgC yr⁻¹) averaged over the 11 TransCom-3 land regions from MsTMIP ("truth") versus the prior biosphere models (NASA-CASA, CASA-GFED, SiB-4 and LPJ) (left column), posterior estimates (middle column) from the OSSE simulations and the corresponding range of prior and posterior NEE (right column). The synthetic observations in these OSSE simulations correspond to the OCO-2 OG observing mode.**