# Peer review of "Prior biosphere model impact on global terrestrial CO2 fluxes estimated"

_Atmospheric Chemistry and Physics, 2018_

## Referee Comment (RC1) · Anonymous Referee #3 · 14 Jun 2019

This paper investigates the impact of prior biospheric CO2 flux models on inverse estimates of terrestrial CO2 fluxes when using synthetic satellite observations. The paper is clear written and well prepared. Only some aspects are still not clear to me, as following:

1 It is concluded that "Overall, even with the availability of dense OCO-2 data, noticeable residual differences (up to ~20-30% globally and 50% regionally) in posterior NEE flux estimates remain that were caused by the choice of prior model flux values and the specification of prior flux uncertainties". From my understanding for inverse problem, if observations contain sufficient information for the target state vector, the results should be, for large part, insensitive to prior. When it strongly depends on prior, either because satellite observations have limited information for flux inversion or the

flux inverse problem is very ill-posed. Could you explain a bit more what is the reason behind?

2 In section 2.4.8, a sanity check has been performed for all the four OSSEs. I am surprised that the check was performed with observational data uncertainty of 0.001%, which is around 0.004 ppm. What assumed is far too unrealistic, current satellite observations can only provide XCO2 observations with uncertainties >= 1.0 ppm. I can understand you do this for sanity check but I am wondering why not give identical inputs (including biosphere model) to all the four OSSEs but under a reasonable observational data uncertainty? You can check if all OSSEs can give similar results which do not have to be the truth. Otherwise, you may still interpret mode-dependent uncertainties as prior dependent uncertainties.

3 Page 10 line 6"The differences between individual model simulations of XCO2 values deviated among themselves by up to $\sim$10 ppm. These large differences in XCO2 values across the four-different prior NEE flux models show that the choice of prior NEE has a large impact on simulated XCO2 values." Even a very strong anthropogenic CO2 source can only introduce a few ppm variations. Are there any explanations for such a large difference?

---

## Author Comment (AC1) · 9 Jul 2019

We thank the reviewer for their valuable comments. Throughout the document reviewer comments are in standard font while author responses are presented with blue text.

This paper investigates the impact of prior biospheric $CO_2$ flux models on inverse estimates of terrestrial $CO_2$ fluxes when using synthetic satellite observations. The paper is clear written and well prepared. Only some aspects are still not clear to me, as following:

1 It is concluded that "Overall, even with the availability of dense OCO-2 data, noticeable residual differences (up to ~20-30% globally and 50% regionally) in posterior NEE flux estimates remain that were caused by the choice of prior model flux values and the specification of prior flux uncertainties". From my understanding for inverse problem, if observations contain sufficient information for the target state vector, the results should be, for large part, insensitive to prior. When it strongly depends on prior, either because satellite observations have limited information for flux inversion or the flux inverse problem is very ill-posed. Could you explain a bit more what is the reason behind?

In this paper, we designed Observing System Simulation Experiments (OSSEs) using synthetic $XCO_2$ retrievals sampled at the OCO-2 satellite spatio-temporal frequency. Figure 2 in the manuscript illustrates the fact that OCO-2 observation density/coverage has noticeable seasonality and incomplete global coverage. For example, during the fall and winter months, minimal OCO-2 observations are obtained in the Northern Boreal regions and observations are limited year-round in the tropical regions of South America and Africa. OCO-2 data is "dense" in comparison with other $CO_2$ observing satellites (e.g., GOSAT) and in situ observations, however, due to cloud coverage and light limitations OCO-2 data coverage is still a limiting factor for a "perfect" inverse model estimate of net ecosystem exchange (NEE). The incomplete global observation coverage of the OCO-2 satellite is a primary reason for the sensitivity of $CO_2$ flux inversions to the assumed prior flux and uncertainty statistics. Previous studies using other observational frameworks also suggest similar sensitivity to prior flux assumptions (e.g., Gurney et al., 2003; Chevallier et al., 2005; Baker et al., 2006, 2010).

2 In section 2.4.8, a sanity check has been performed for all the four OSSEs. I am surprised that the check was performed with observational data uncertainty of 0.001%, which is around 0.004 ppm. What assumed is far too unrealistic, current satellite observations can only provide $XCO_2$ observations with uncertainties >= 1.0 ppm. I can understand you do this for sanity check but I am wondering why not give identical inputs (including biosphere model) to all the four OSSEs but under a reasonable observational data uncertainty? You can check if all OSSEs can give similar results which do not have to be the truth. Otherwise, you may still interpret mode-dependent uncertainties as prior dependent uncertainties.

As noted correctly by the reviewer, these "pseudo" experiments were 'sanity checks' for demonstrating the robustness of the inversion setup using synthetic OCO-2 data. We applied the very low uncertainty values to our synthetic OCO-2 observations (0.001%), using all four variable prior flux estimates, in

order to test two aspects of the inversion: 1) spread in posteriors (uncertainty) and 2) ability to reproduce the "truth" (accuracy). Both of these are important when testing/demonstrating the robustness of the inversion system. In order to eliminate the impact of observational uncertainty, we apply this very small uncertainty value which is justified in the OSSE framework since the synthetic observations are produced with known $CO_2$ fluxes and transport. If we set observational uncertainty to realistic values (such as ~1.0 ppm) in these test simulations we would systematically get posterior NEE spread which is not specifically caused by variable priors, but a combination of variable priors and observational uncertainty. It should be noted that we do use more realistic observation uncertainty for OCO-2 data in this study for OSSEs outside of the "pseudo" simulations.

We agree with the reviewer that if we ran the four OSSEs, with the four different prior fluxes, applying a realistic observational error we could compare the spread of the posteriors, however, would inherently not fully reproduce the truth due to the lowered constraint by observations. Therefore, this setup would only test one aspect of the system (uncertainty) while the current "pseudo" simulations test both accuracy and uncertainty.

3 Page 10 line 6"The differences between individual model simulations of $XCO_2$ values deviated among themselves by up to ~10 ppm. These large differences in $XCO_2$ values across the four-different prior NEE flux models show that the choice of prior NEE has a large impact on simulated $XCO_2$ values." Even a very strong anthropogenic $CO_2$ source can only introduce a few ppm variations. Are there any explanations for such a large difference?

We agree with the reviewer that variations in atmospheric $CO_2$ concentrations due to strong anthropogenic sources are only on the order of a few ppm. However, global atmospheric $CO_2$ concentrations are primarily controlled by biospheric $CO_2$ fluxes (e.g., Fung et al., 1983, Schimel et al., 2001, Le Quéré et al., 2018). On a global scale, anthropogenic annual $CO_2$ fluxes (~10 PgC yr$^{-1}$) are far less compared with natural biospheric $CO_2$ fluxes (~250 PgC yr$^{-1}$, sum of the absolute value of photosynthetic uptake and respiration) (Ciais et al., 2013). Furthermore, biospheric $CO_2$ fluxes are among the most uncertain components of the carbon cycle (Huntzinger et al., 2012, Schimel et al., 2015, Zscheischler et al., 2017). Recent studies have shown that global fossil fuel emissions have an estimated uncertainty of < 10% (e.g., Le Quéré et al., 2018) while we show from the Multiscale Synthesis and Terrestrial Model Intercomparison Project (MsTMIP) biospheric NEE model ensemble (Huntzinger et al., 2013; 2018, Fisher et al., 2016a; 2016b) that global land NEE uncertainties are as high as ~50-70%. Overall, due to the large impact of NEE on atmospheric $CO_2$ concentrations, and the significant difference between prior NEE estimates (i.e., large uncertainty), it is expected that $XCO_2$ values would be noticeably different when changing the prior NEE flux.

**References**

[revised manuscript text omitted]

---

## Referee Comment (RC2) · Anonymous Referee #4 · 20 Aug 2019

Review of "Prior biosphere model impact on global terrestrial CO2 fluxes estimated from OCO-2 retrievals" from Sajeev Philip et al

This work details a set of pseudo-data experiments in which the prior flux and prior flux uncertainty are varied in a systematic manner in order to assess the sensitivity of atmospheric inversions with satellite data to the prior flux constraint. The manner in which this is carried out is a good template for assessing sensitivity to inversion system ingredients.

General comments: 1. The presentation in terms of images makes some of the conclusions and discussion difficult to assess. Specifically I'm referring to Figures 3-5, where the "meat" of the results is contained. I would suggest that the NEE range differences are great material for a table, and that the other two columns are perfect to be

condensed into stacked bars or something similar for ease of visibility.

2. My main complaint with this is that there are a few papers out there now that use ensembles of models for inference (Basu et al, 2018; Crowell et al, 2019; older ones like Peylin et al, 2013), and that this would be a much more effective paper if it were to place itself in the context of the "uncertainty budget" for these other papers. For example, Crowell et al (2019) presents results in the form of ensemble means and standard deviations, and Basau et al (2018) presents ensemble members individually, and the results here could be placed beside the Basu et al (2018) results to attempt to explain the scatter in the flux results in Crowell et al (2019). That sort of analysis would elevate the messages in this paper to a greater impact.

Specific comments: Page 4, Line 28 - how much do the models really vary using this approach? I'd guess not much. Can you provide a figure at a well known flux tower site as a demonstration?

Page 7, Line 16-18 Chevallier et al also went after this by looking at structural uncertainties in the ORCHIDEE ecosystem model. Chevallier, F., Viovy, N., Reichstein, M., & Ciais, P. (2006). On the assignment of prior errors in Bayesian inversions of CO 2 surface fluxes. Geophysical Research Letters, 33(13), L13802. https://doi.org/10.1029/2006GL026496

Page 8, Lines 5-9 - these errors are even smaller than those predicted by the Level 2 retrieval, and those are known to be underestimated (from various uncertainty quantification talks). I wonder, could this really be a sign that your prior errors need tuning, rather than your observation errors? In the OSSE setting, it's equivalent which way you go, but in real data settings, this choice can matter a lot. There are other metrics to optimize the prior errors vs. the observational error statistics, such as the Desrozier approach (commonly used in numerical weather prediction)

Page 10, Lines 34-35 - this is often called the "uncertainty reduction", assuming the standard deviation is a proxy for the uncertainty

---

## Editor Comment (EC1) · Joshua Fu (Editor) · 20 Aug 2019

The comments posted below is on a referee's behalf:

\*\*\*\*\*\*\*\*\*\*\*\*\*\*\*\*\*\*\*\*\*\*\*\*\*\*\*\*\*\*\*

The investigators have conducted a series of OSSEs to assess the impact of prior biospheric fluxes on posterior fluxes in CO2 flux inversions using OCO-2 data. The lack of robust regional flux estimates is a major issue in the flux inversion community. It is known that the choice of prior fluxes can impact posterior flux estimates, contributing to discrepancies between different inversions. This study has conducted the most detailed and thorough sensitivity analysis to date to quantify the potential impact of the prior fluxes in CO2 inversions. The manuscript is well written and I recommend it

for publication in ACP with minor revisions to address my mostly technical comments below.

Comments

1. Page 4, line 3-4: Change "NEE flux (balanced biosphere)" to just "NEE flux".

2. Page 4, line 28: How different is the diurnal variation between the truth and the prior models? This information could be included in the Supplement.

3. Page 6, line 2: Change "CO2 at August" to "CO2 on August".

4. Page 6, line 13: OCO-2 XCO2 is not actually retrieved using Equation (1). Rather, the retrieval is expressed as Equation (1), after the fact.

5. Page 6, line 16: Please add "column" between "a" and "averaging" kernel.

6. Page 6, lines 32, 34, 35, etc...: Please change "model grid" to "model grid box" when discussing the model grid boxes. For example, on lines 34-35 it should read "the jth model grid box" instead of the "jth model grid".

7. Page 7, line 5, and page 8, line 31: Same comment as above regarding the "model grid" vs "model grid box".

8. Page 7, Equation (3): Shouldn't this equation be similar to Equation (1) since the observation operator is transforming the model into the observation space? For example, the "ya" and "Ma" in this equation should be the same as "ca" and "XCO2a" used in Equation (1), respectively. The only quantity that should be different in this expression is "f(x)", which represents the simulated profile.

9. Page 7, line 37: Something is missing between "Similar" and "to prior error statistics". Should this say "Similar to our treatment of the prior error statistics. . ."

10. Page 8, lines 20: Figure S2 is useful for the reader who is unfamiliar with the TransCom domains. Furthermore, it has the numerical ordering of the regions that is

useful for interpreting Table 3. I would suggest moving this into the main manuscript.

11. Page 10: Figures S5 and S7 show the spatial distribution of the results and complements the information shown in Figures 3-5. I would suggest moving Figures S5 and S7 in the main section of the manuscript, which currently has only five figures.

12. Page 11, lines 31-33: I don't understand the statement here that the NEE estimates are more sensitive to the prior error when there are sufficient observations available and large differences between the truth and prior. Is this due to the inversion approach used here? Is it because the prior error is a relative error so when the flux is larger, the error is also larger, which gives the inversion more flexibility in adjusting the fluxes?

---

## Author Comment (AC3) · 14 Sep 2019

**Response to Anonymous Referee (through Editor's comments)**

We thank the editor for adding these addition valuable reviewer comments. Throughout the document reviewer comments are displayed in standard font while author responses are presented with blue text.

The investigators have conducted a series of OSSEs to assess the impact of prior biospheric fluxes on posterior fluxes in $CO_2$ flux inversions using OCO-2 data. The lack of robust regional flux estimates is a major issue in the flux inversion community. It is known that the choice of prior fluxes can impact posterior flux estimates, contributing to discrepancies between different inversions. This study has conducted the most detailed and thorough sensitivity analysis to date to quantify the potential impact of the prior fluxes in $CO_2$ inversions. The manuscript is well written and I recommend it for publication in ACP with minor revisions to address my mostly technical comments below.

Comments

1. Page 4, line 3-4: Change "NEE flux (balanced biosphere)" to just "NEE flux".

We have removed the following text "(balanced biosphere for the 1998-2017 time period)" in the revised manuscript.

2. Page 4, line 28: How different is the diurnal variation between the truth and the prior models? This information could be included in the Supplement.

A similar comment was made by Referee #4. In order to demonstrate the difference between the "true" NEE and prior model data in regards to diurnal variability, monthly-averaged 3-hourly (MsTMIP) and hourly (for the 4 prior biosphere models) NEE values were plotted for July 2015 at the well-known Park Falls flux tower site (45.95°N, 90.27°W). This figure has been added to the revised manuscript in the supplementary section as Fig. S1. We added additional text to Sect. 2.1 of the revised manuscript which reads: "We allow the "true" and prior models to have different diurnal variability in order to represent a realistic scenario, as prior models will differ some from the actual diurnal variability of NEE in nature. In general, the diurnal variability of NEE is similar between the "true" and individual prior models. An example is shown in Fig. S1 where it can be seen that the diurnal NEE from the "true" and prior models for July 2015 at the Park Falls flux tower site (45.95°N, 90.27°W) have near identical temporal diurnal patterns and only differ in NEE magnitude.".

3. Page 6, line 2: Change "CO$_2$ at August" to "CO$_2$ on August".

This has been corrected.

4. Page 6, line 13: OCO-2 XCO2 is not actually retrieved using Equation (1). Rather, the retrieval is expressed as Equation (1), after the fact.

The following text "OCO-2 XCO2 is retrieved using the following equation" has been changed to "The retrieval of OCO-2 XCO2 is expressed as Eq. (1)" in the revised manuscript.

5. Page 6, line 16: Please add "column" between "a" and "averaging" kernel.

This has been corrected.

6. Page 6, lines 32, 34, 35, etc...: Please change "model grid" to "model grid box" when discussing the model grid boxes. For example, on lines 34-35 it should read "the j$^{th}$ model grid box" instead of the "j$^{th}$ model grid".

This has been corrected.

7. Page 7, line 5, and page 8, line 31: Same comment as above regarding the "model grid" vs "model grid box".

This has been corrected

8. Page 7, Equation (3): Shouldn't this equation be similar to Equation (1) since the observation operator is transforming the model into the observation space? For example, the "ya" and "Ma" in this equation should be the same as "ca" and "XCO2a" used in Equation (1), respectively. The only quantity that should be different in this expression is "f(x)", which represents the simulated profile.

We thank the reviewer for recognizing this. In the updated manuscript we now use the same symbols for the prior CO$_2$ profile and column CO$_2$ values from OCO-2 data in the retrieval (Eq. (1)) and observation operator (Eq. (3)) description.

9. Page 7, line 37: Something is missing between "Similar" and "to prior error statistics". Should this say "Similar to our treatment of the prior error statistics:"

This text in the revised manuscript has been corrected to "Similar to the treatment of prior error statistics".

10. Page 8, lines 20: Figure S2 is useful for the reader who is unfamiliar with the TransCom domains. Furthermore, it has the numerical ordering of the regions that is useful for interpreting Table 3. I would suggest moving this into the main manuscript.

We thank the reviewer for this suggestion and we have moved this figure from the supplementary material to be Fig. 1 in the main manuscript.

11. Page 10: Figures S5 and S7 show the spatial distribution of the results and complements the information shown in Figures 3-5. I would suggest moving Figures S5 and S7 in the main section of the manuscript, which currently has only five figures.

We thank the reviewer for this suggestion and we have moved Fig. S5 and S7 from the supplementary material to the main manuscript and are now Fig. 3 and 5, respectively.

12. Page 11, lines 31-33: I don't understand the statement here that the NEE estimates are more sensitive to the prior error when there are sufficient observations available and large differences between the truth and prior. Is this due to the inversion approach used here? Is it because the prior error is a relative error so when the flux is larger, the error is also larger, which gives the inversion more flexibility in adjusting the fluxes?

We refer the reviewer to the end of Sect. 2.4.5 where the discussion of the prior error calculations is presented. In this study, we calculate prior error values from the standard deviation (SD) of the four prior models (as a NEE magnitude) which is then divided by the magnitude of the NEE flux from each individual model (fractional error). We did this in order for our relative error values (fractional error) to be representative of absolute error magnitudes as defined by the SD calculation. This is done so large prior error can be applied to small fluxes and vice versa.

The reason that posterior NEE estimates are more sensitivity to prior error values in regions with sufficient observational coverage/density and large differences in "true" and prior fluxes is as follows. When the model has sufficient observations to constrain NEE fluxes, and the "truth" and prior are noticeably different, the model must optimize the prior flux significantly to match the truth. When the prior error is too small (e.g., 10% in all grid boxes as we apply in our sensitivity study), the model will not have enough freedom to deviate from the prior estimate to match the truth. For cases when large prior error is assigned (e.g., 100% in all grid

boxes), the model is able to diverge greatly from the prior in all regions in order to match observations. As demonstrated in Sect. 3.3 of the manuscript, when applying different prior NEE estimates, the model will optimize $CO_2$ fluxes in variable ways to match atmospheric observations. The larger the prior error applied to each prior flux will add additional flexibility in each simulation allowing the model to match the atmospheric observations in increasingly different ways when using variable prior fluxes.

---

## Author Response (AR1)

**Response to Anonymous Referee #4**

We thank the reviewer for their valuable comments. Throughout the document reviewer comments are displayed in standard font while author responses are presented with blue text.

This work details a set of pseudo-data experiments in which the prior flux and prior flux uncertainty are varied in a systematic manner in order to assess the sensitivity of atmospheric inversions with satellite data to the prior flux constraint. The manner in which this is carried out is a good template for assessing sensitivity to inversion system ingredients.

General comments: 1. The presentation in terms of images makes some of the conclusions and discussion difficult to assess. Specifically I'm referring to Figures 3-5, where the "meat" of the results is contained. I would suggest that the NEE range differences are great material for a table, and that the other two columns are perfect to be condensed into stacked bars or something similar for ease of visibility.

We appreciate the comment from the reviewer regarding Fig. 3-5 in the original manuscript (now Fig. 6-8 of the updated version of the manuscript). While producing the manuscript we spent significant time and effort developing these figures in order to present as much information as possible in a clear manner. Due to the large amount of information provided in these figures, it proved a difficult task. We tested numerous figure types (line graphs, stacked bar graphs, pie charts, etc.) and determined the current layout was the most effective. Specifically, for the stacked bar charts, when multiple prior or posterior NEE estimates are very similar (which happens frequently) for a specific region/season, they will not be visible as there is not enough difference in the values. After numerous versions of Fig. 3-5 we reverted back to the grouped bar chart format as the best way to display the main results of this study.

We point the reviewer to Table 3 of the manuscript where NEE range values illustrated in Fig. 6 are already listed (along with "true", multi-model prior and posterior means, and standard deviations of NEE values shown in Fig. 6). We have added the following text to the Fig. 6 caption to guide the reader to this information: "Detailed statistics of the "truth", multi-model means of prior and posterior NEE estimates, standard deviations, and ranges displayed in this figure are listed in Table 3.". In response to the reviewer, we have added Table S1 and S2 to the supplementary material of the revised manuscript in order to provide the "truth", mean, standard deviation, and range of the NEE values calculated in the sensitivity studies of prior error statistics and ocean glint (OG) simulations, respectively (detailing Fig 7 and 8).

2. My main complaint with this is that there are a few papers out there now that use ensembles of models for inference (Basu et al, 2018; Crowell et al, 2019; older ones like Peylin et al, 2013), and that this would be a much more effective paper if it were to place itself in the context of the "uncertainty budget" for these other papers. For example, Crowell et al (2019) presents results in the form of ensemble means and standard deviations, and Basau et al (2018) presents ensemble members individually, and the results here could be placed beside the Basau et al (2018) results to attempt to explain the scatter in the flux results in Crowell et al (2019). That sort of analysis would elevate the messages in this paper to a greater impact.

We fully agree with the reviewer that the comparison of our results with Crowell et al. (2019) and Basu et al. (2018) is important and will greatly improve the impact of this study. We thought of this prior to submission of our manuscript, however, Crowell et al. (2019) (along with the OCO-2 Multi-model Inter-comparison Project (MIP) Level 4 (L4) $CO_2$ flux data) and the supplementary tables of Basu et al. (2018) were not yet published. However, now that this information is available, we have performed the suggested comparison and have added the following text to the conclusions section of the revised manuscript:

"As explained earlier in this study, estimates of surface $CO_2$ fluxes from numerous inversion systems in the OCO-2 MIP ensemble model framework, using identical OCO-2 observations, result in different optimized/posterior regional NEE fluxes (Crowell et al., 2019). This inverse model variance can be due to numerous factors (e.g., model transport, inversion methods, observation errors, etc.) including prior model mean and uncertainty estimates. In order to estimate the amount of variance in the results of posterior NEE values from the OCO-2 MIP which could be due to prior flux estimates, we compare our OSSE derived residual posterior NEE range (using LN+LG) to the range in the posterior Level-4 OCO-2 Flux data (using both LN and LG) (https://www.esrl.noaa.gov/gmd/ccgg/OCO2/index.php) in each TransCom-3 region. This comparison suggests that prior NEE and uncertainty statistics could contribute 10-30% (average ~20%) of annually-averaged NEE variance calculated for each TransCom-3 region in the OCO-2 Level-4 MIP flux data. Comparing this contribution of prior model impact to the OSSE study by Basu et al. (2018), which calculated the impact of atmospheric transport on posterior NEE estimates when assimilating OCO-2 observations, this contribution is ~50% less compared to the impact of atmospheric transport. From our study and Basu et al. (2018) it is estimated that the combination of prior flux/uncertainty assumptions and atmospheric transport could contribute on average ~50% of the annually-averaged posterior NEE variance of the OCO-2 MIP study."

Also, the references to "Crowell et al. in prep." have been updated to "Crowell et al., 2019".

Specific comments: Page 4, Line 28 - how much do the models really vary using this approach? I'd guess not much. Can you provide a figure at a well known flux tower site as a demonstration?

A similar comment was made by the editor on behalf of an additional referee. In order to demonstrate the difference between the "true" NEE and prior model data in regards to diurnal variability, monthly-averaged 3-hourly (MsTMIP) and hourly (for the 4 prior biosphere models) NEE values were plotted for July 2015 at the well-known Park Falls flux tower site (45.95°N, 90.27°W). This figure has been added to the revised manuscript in the supplementary section as Fig. S1. We added additional text to Sect. 2.1 of the revised manuscript which reads: "We allow the "true" and prior models to have different diurnal variability in order to represent a realistic scenario, as prior models will differ some from the actual diurnal variability of NEE in nature. In general, the diurnal variability of NEE is similar between the "true" and individual prior models. An example is shown in Fig. S1 where it can be seen that the diurnal NEE from the "true" and prior models for July 2015 at the Park Falls flux tower site (45.95°N, 90.27°W) have near identical temporal diurnal patterns and only differ in NEE magnitude.".

Page 7, Line 16-18 Chevallier et al also went after this by looking at structural uncertainties in the ORCHIDEE ecosystem model. Chevallier, F., Viovy, N., Reichstein, M., & Ciais, P. (2006). On the assignment of prior errors in Bayesian inversions of $CO_2$ surface fluxes. Geophysical Research Letters, 33(13), L13802. https://doi.org/10.1029/2006GL026496

We appreciate the reviewer pointing out this study. We have added this reference to the revised manuscript and additional text in Sect. 2.4.5 which reads: "…, using continuous in situ measurements of $CO_2$ flux compared to model simulations to inform prior errors (Chevallier et al., 2006), …"

Page 8, Lines 5-9 - these errors are even smaller than those predicted by the Level 2 retrieval, and those are known to be underestimated (from various uncertainty quantification talks). I wonder, could this really be a sign that your prior errors need tuning, rather than your observation errors? In the OSSE setting, it's equivalent which way you go, but in real data settings, this choice can matter a lot. There are other metrics to optimize the prior errors vs. the observational error statistics, such as the Desrozier approach (commonly used in numerical weather prediction)

We agree with the reviewer that the observational error values applied to our OSSE simulations for synthetic OCO-2 data are smaller than that in the real data. It should be noted that the simulated synthetic OCO-2 data produced in this study is done so using known fluxes and transport and sampling a model-predicted atmosphere, thus errors should be small. This manner of producing synthetic satellite observations makes it so there is no model-data mismatch or systematic error which is the major fraction of OCO-2 error values.

The reviewer is correct in the fact that the manner in which observational and prior error are adjusted is equivalent for OSSE simulations (such as this work), however, can impact the results of posterior estimates in "real" inverse model simulations. In addition to the fact we use model produced observations in our OSSEs, we also decided not to reduce our prior error statistics in order to be representative of how prior error is typically calculated/treated in "real" data assimilations. Calculating the difference or spread between state-of-the-science model ensemble members as the best estimate of our knowledge of a process is commonly done to define prior error values. Furthermore, if prior error values are reduced and observation error remain large, this results in an inversion system which is limited in the ability to deviate from the prior and posterior NEE spread calculated would be due to both observational error and prior NEE flux/error. Therefore, by reducing observational error (attached to data which have no model-data mismatch or systematic error in our OSSE framework), and defining prior error as is done in "real" inversions, provides a true representation of the spread in posterior estimates primarily due to prior flux/error.

Page 10, Lines 34-35 - this is often called the "uncertainty reduction", assuming the standard deviation is a proxy for the uncertainty.

We have modified the following text in Sect. 3.4 of the revised manuscript to demonstrate this: "The reduction in the SD of NEE (also known as uncertainty reduction) in most regions/seasons, calculated as $100 \times (1 - \text{(posterior NEE SD)}/\text{(prior NEE SD)})$ is generally > 70% and up to 98%".

We refer the reviewer to the end of Sect. 2.4.5 where the discussion of the prior error calculations is presented. In this study, we calculate prior error values from the standard deviation (SD) of the four prior models (as a NEE magnitude) which is then divided by the magnitude of the NEE flux from each individual model (fractional error). We did this in order for our relative error values (fractional error) to be representative of absolute error magnitudes as defined by the SD calculation. This is done so large prior error can be applied to small fluxes and vice versa.

The reason that posterior NEE estimates are more sensitivity to prior error values in regions with sufficient observational coverage/density and large differences in "true" and prior fluxes is as follows. When the model has sufficient observations to constrain NEE fluxes, and the "truth" and prior are noticeably different, the model must optimize the prior flux significantly to match the truth. When the prior error is too small (e.g., 10% in all grid boxes as we apply in our sensitivity study), the model will not have enough freedom to deviate from the prior estimate to match the truth. For cases when large prior error is assigned (e.g., 100% in all grid boxes), the model is able to diverge greatly from the prior in all regions in order to match observations. As demonstrated in Sect. 3.3 of the manuscript, when applying different prior NEE estimates, the model will optimize $CO_2$ fluxes in variable ways to match atmospheric observations. The larger the prior error applied to each prior flux will add additional flexibility in each simulation allowing the model to match the atmospheric observations in increasingly different ways when using variable prior fluxes.

[revised manuscript text omitted]

*TransCom-3 region name and location displayed in Fig. 1.

[Figure]

Figure 1: The TransCom-3 land region boundaries used to aggregate $CO_2$ fluxes for evaluation.

[Figure]

Figure 2: Prior multi-model (NASA-CASA, CASA-GFED, SiB-4 and LPJ biosphere models) seasonally-averaged NEE (gC m$^{-2}$ day$^{-1}$) (left column) and NEE standard deviation (gC m$^{-2}$ day$^{-1}$) (right column) for the year 2015.

[Figure]

Figure 3: Seasonally-averaged NEE range (gC m$^{-2}$ day$^{-1}$) of the four prior biosphere models (NASA-CASA, CASA-GFED, SiB-4 and LPJ) (left) and posterior estimates (right) from the OSSE simulations. The synthetic observations in these OSSE simulations correspond to the OCO-2 LN+LG observing modes.

[Figure]

**Figure 4: Total number of OCO-2 LN and LG XCO₂ observations (left column) and the corresponding seasonally-averaged multi-model (NASA-CASA, CASA-GFED, SiB-4 and LPJ biosphere models) mean GEOS-Chem-simulated prior XCO₂ (ppm) (right column) in 2015.**

[Figure]

Figure 5: Seasonally-averaged XCO2 range (ppm) from GEOS-Chem forward model simulations using the four prior biosphere models (NASA-CASA, CASA-GFED, SiB-4 and LPJ) (left) and the corresponding posterior estimates (right) from the OSSE simulations. The synthetic observations in these OSSE simulations correspond to the OCO-2 LN+LG observing modes.

[Figure]

**Figure 6: Seasonally-averaged NEE (PgC yr⁻¹) averaged over the 11 TransCom-3 land regions from MsTMIP ("truth") versus the prior biosphere models (NASA-CASA, CASA-GFED, SiB-4 and LPJ) (left column), posterior estimates (middle column) from the OSSE simulations and the corresponding range of prior and posterior NEE estimates (right column). The synthetic observations in these OSSE simulations correspond to the OCO-2 LN+LG observing modes. Detailed statistics of the "truth", multi-model means of prior and posterior NEE estimates, standard deviations, and ranges displayed in this figure are listed in Table 3.**

[Figure]

**Figure 7: Seasonally-averaged NEE (PgC yr⁻¹) averaged over the 11 TransCom-3 land regions from MsTMIP ("truth") versus CASA-GFED prior biosphere model (left column), posterior estimates with the three different prior uncertainties (middle column) and the corresponding range of posterior NEE (right column). The synthetic observations in OSSE simulations correspond to the OCO-2 LN+LG observing modes. Detailed statistics of the "truth", prior, multi-model mean of posterior NEE estimates, standard deviations, and ranges displayed in this figure are listed in Table S1.**

[Figure]

**Figure 8: Seasonally-averaged NEE (PgC yr⁻¹) averaged over the 11 TransCom-3 land regions from MsTMIP ("truth") versus the prior**
**biosphere models (NASA-CASA, CASA-GFED, SiB-4 and LPJ) (left column), posterior estimates (middle column) from the OSSE**
**simulations and the corresponding range of prior and posterior NEE (right column). The synthetic observations in these OSSE**
**simulations correspond to the OCO-2 OG observing mode. Detailed statistics of the "truth", multi-model means of prior and posterior**
**NEE estimates, standard deviations, and ranges displayed in this figure are listed in Table S2.**